# Indications of Induction and Caesarean Sections Performed Using the Robson Classification in a University Hospital in Spain from 2010 to 2021

**DOI:** 10.3390/healthcare11111521

**Published:** 2023-05-23

**Authors:** Rafael Vila-Candel, Nadia Piquer-Martín, Nerea Perdomo-Ugarte, José Antonio Quesada, Ramón Escuriet, Anna Martin-Arribas

**Affiliations:** 1Department of Nursing, Universitat de València, 46007 Valencia, Spain; 2Department of Obstetrics and Gynaecology, Hospital Universitario de la Ribera, 46600 Valencia, Spain; 3Foundation for the Promotion of Health and Biomedical Research in the Valencian Region (FISABIO-SP), 46020 Valencia, Spain; 4Department of Clinical Medicine, Universidad Miguel Hernández, 03202 Elche, Spain; 5Network for Research on Chronicity, Primary Care and Health Promotion (RICAPPS), 03550 Alicante, Spain; 6School of Health Sciences Blanquerna, Universitat Ramon Llull, C/Padilla 326, 08025 Barcelona, Spain

**Keywords:** Robson classification, caesarean section, labour induction, indications, onset of labour, mode of birth

## Abstract

*Background:* The Robson Ten Group Classification System (RTGCS) enables the assessment, monitoring, and comparison of caesarean section rates both within healthcare facilities and between them, and the indications of caesarean sections (CS) performed in a maternity ward. The aims of the present study were to conduct an analysis to assess the levels and distribution of birth from a descriptive approach by CS in La Ribera University Hospital (Spain) between 2010–2021 using the Robson classification; to describe the indications for the induction of labour and the causes of caesarean sections performed; and to examine the association between the induction of labour and CS birth. *Methods*: A retrospective study between 1 January 2010 and 31 December 2021. All eligible women were classified according to the RTGCS to determine the absolute and relative contribution by each group to the overall CS rate. The odds ratio (OR) of the variables of interest was estimated by logistic regression. In an analysis of the subgroups, the level of significance was adjusted using the Bonferroni method. *Results:* 20,578 women gave birth during the study period, 19% of them by CS. In 33% of births, induction was performed, and the most common cause was the premature rupture of membranes. Group 2 (nulliparous with induced labour/elective CS before labour) accounted for the largest contribution to the overall rate of CS (31.5%) and showed an upward trend from 23.2% to 39.7% in the time series, increasing the CS rate by 6.7%. The leading cause of CS was suspected fetal distress, followed by induction failure. *Conclusions:* In our study, Robson Group 2 was identified as the main contributor to the hospital’s overall CS rate. Determining the causes of induction and CS in a population sample classified using the RTGCS enables the identification of the groups with the greatest deviation from the optimal rate of CS and the establishment of improvement plans to reduce the overall rate of caesarean sections in the maternity unit.

## 1. Introduction

The worldwide rise in caesarean section (CS) rates is a major public health concern and a cause of considerable debate due to their steady increase, the lack of consensus on the appropriate CS rate, the associated short- and long-term maternal and neonatal risks and costs, and the inequity in access [1,2]. In Europe, the rate of CS varies considerably between countries from 15% to 45%, and the reasons for this situation appear to be multiple, complex, and, in many cases, country specific [3]. 

The Robson classification system is a tool used to classify, monitor, and compare CS rates in a standardized, reliable, consistent, and action-oriented manner, with the aim of understanding the drivers and contributors of this trend across different settings and populations and identifying areas where interventions may be needed to reduce CS rates [4,5,6]. 

In 2017, the World Health Organization (WHO) developed guidelines for its use, implementation, and interpretation, including the standardization of terms and definitions, in order to help healthcare facilities adopt and use the Robson classification [7]. In 2022, the WHO has implemented another tool to facilitate the use of the Robson classification, the “Robson Platform” [8]. This online platform allows the continuous monitoring of caesarean sections using the Robson Ten Group Classification System (RTGCS). Data is openly available and is updated in real time as soon as new data is uploaded by the maternity healthcare facilities. Given the constant increase in CS use globally, the substantial inequalities in low- and middle-income countries, and the potential impact on perinatal health outcomes, continuous monitoring of caesarean sections using such tools should be established as a global monitoring priority [9,10]. 

Furthermore, over the last decades, rates of induction of labour have also increased from twofold to fourfold in high-income countries, and the WHO estimates that 25% of women undergo induced labour in this context [11]. Previous studies [12,13] have associated the induction of labour with negative perinatal outcomes, such as a greater number of admissions to the neonatal intensive care unit, chorioamnionitis, postpartum haemorrhage, or perineal injuries, among other relevant outcomes. In addition, according to the studies [14,15], the risk of a CS in women that undergo induced labour is between two and three times higher when compared to women that have a spontaneous onset of labour, with failed induction of labour being the most frequent cause for the indication of a CS. In contrast, other authors have observed that inductions performed at week 39 in nulliparous pregnant women have reduced the number of CS [16,17].

The aims of the present study were to conduct an analysis to identify the groups of women that contribute most and least to overall CS rates, assess levels and distribution of births by CS from a descriptive approach in La Ribera University Hospital (Spain) between 2010–2021 using the Robson classification, describe the indications for inducing labour and the caesarean sections performed, and examine the association between induced labour and CS birth.

## 2. Materials and Methods

### 2.1. Design, Population, and Sample

This is a retrospective observational study of all births attended at Hospital Universitario de la Ribera (HULR) between 1 January 2010 and 31 December 2021. Data were obtained by the research team from the review of the electronic medical record of each of the cases included. 

Currently, HULR attends an average of 1300 births per year and has a potential reference population of 250,000 inhabitants. Miscarriages at less than 22 weeks of gestation or birth weight under 500 g were considered exclusion criteria.

We obtained ethical clearance from the Research Ethics Committee of Hospital de la Ribera. Confidential data and complete information concerning participants were secured throughout the review process. Due to the nature of the retrospective study, neither the patient information sheet nor their informed consent were necessary, because only the electronic records were analysed, and there was no contact with any participant.

### 2.2. Data Collection Tools

The Robson ten group classification system (RTGCS) was used to categorize caesarean sections in the selected sample. Table 1 sets out the definitions of each group.

The systems department provided us with the births attended in the study period, along with the variables necessary for the tabulation of the RTGCS. The five variables for the RTGCS were collected, which included obstetric history (parity and previous CS), type of onset of labour (spontaneous, induced, or CS before labour), fetal presentation (cephalic, breech, or transverse), and number of newborns and gestational age (preterm or full term). In addition, sociodemographic variables (country of origin, age), obstetric-perinatal variables (sex of the newborn, birth weight, indication of induction, and cause of indication of CS) were included. Finally, various variables were categorized to obtain an analysis of perinatal outcomes, such as low birth weight (birth weight < 2500 g), small for gestational age (birth weight lower than 10th percentile for gestational age) [18], suspected intrauterine growth restriction (birth weight lower than 5th percentile for gestational age) [18], or macrosomia (birth weight > 4000 g) and preeclampsia (yes/no). Next, the research team grouped all records in which the onset of labour was an induction to examine the indications. Finally, all cases where birth resulted in caesarean section were grouped together to determine the indications.

### 2.3. Statistical Analysis

The basic descriptive methods of calculation of mean and standard deviation were used for continuous variables, median and interquartile range for nonnormal distributions, and absolute and relative frequencies for categorical variables. The Kolmogorov–Smirnov goodness-of-fit test was used to assess whether the variables complied with the principle of normality. The associations between the qualitative variables were analysed by means of contingency tables, applying the Chi-Square test. For quantitative variables, mean values were compared using Student’s *t*-test or the ANOVA procedure or the Kruskal–Wallis test. The odds ratio (OR) and 95% CI of the variables of interest was estimated by simple logistic regression. In the analysis of the subgroups, the level of significance was adjusted using the Bonferroni method. The level of statistical significance defined was *p* < 0.05. Data were analysed using Statistical Package for the Social Version 28.0.1 Sciences (SPSS Inc., Chicago, IL, USA).

## 3. Results

A total of 20,578 births attended at HULR during the study period were analysed. The women’s mean age was 30.85 ± 5.77 years—79.4% were of Spanish origin and 54.9% were primiparous. A total of 62% had spontaneous onset of labour, and 32.9% were induced. The mean birth weight was 3290.39 ± 473.35 g, 48.6% female, and 51.4% male. The CS rate was 19.0%, with 14.5% corresponding to the rate of intrapartum CS and 4.5% to elective caesarean sections, respectively. Table 2 shows the distribution of the type of onset of labour and the mode of birth of the sample analysed.

To determine whether there was a percentage change in the variables analysed in the time series, we categorized the distribution into three different periods (2010–2013; 2014–2017 and 2018–2021). Table 3 presents the distribution in socio-demographic and obstetric variables among women at HULR, a hospital in Spain, between 2010 and 2021. The percentage of Spanish women decreased from 83.0% in 2010-2013 to 70.9% in 2018–2021, while the percentage of foreign women increased from 17.0% to 29.1% during the same period (*p* < 0.001). The proportion of male and female babies remained stable across the three periods with no significant difference. Most women had singleton pregnancies, and the proportion of multiple fetuses remained low and stable. The percentage of women with one previous pregnancy increased from 55.6% in 2010–2013 to 61.0% in 2018–2021, while the percentage of women with two or more previous pregnancies decreased from 44.4% to 36.1% over the same period (*p* < 0.001). The mean maternal age increased from 30.4 ± 5.5 years in 2010–2013 to 31.2 ± 6.1 years in 2018–2021, with a statistically significant difference between the three periods (*p* < 0.001). The mean number of births per woman decreased from 0.6 in 2010–2013 to 0.5 in 2014–2017, but then increased back to 0.6 in 2018–2021, with a statistically significant difference between the three periods (*p* < 0.001). The mean birth weight remained relatively stable over the years, with a statistically significant difference between the three periods (*p* = 0.022), although the differences were small (23 g).

Similarly, we examined socio-demographic factors such as country of origin and maternal age, as well as perinatal variables such as sex of newborn and birth weight, for each subgroup across the time series (refer to Appendix A). Our analysis revealed statistically significant differences in country of origin for subgroups 1, 2, 3, 6, 8, and 10, with Spanish women being the most prevalent (*p* < 0.001; *p* < 0.001; *p* < 0.001; *p* < 0.001; *p* = 0.013; *p* = 0.023, respectively). Maternal age increased over the course of the time series, with significant differences observed in groups 1, 2, 3, 4, 5, 7, and 10, particularly in groups 5 and 7 (*p* = 0.005; *p* = 0.012; *p* < 0.001; *p* < 0.001; *p* < 0.001; *p* < 0.001; *p* = 0.041, respectively), resulting in an average maternal age increase from 31 to 35 years for these last two groups. Lastly, we observed that statistically significant differences were present in birth weight for subgroup 3 (*p* < 0.001), with a difference of 51g between 2010–2013 and 2018–2021.

Table 4 shows the distribution of the births carried out in the study period, taking the RTGCS into account. Overall, Groups 1 to 4 account for the largest population size.

We were interested in analysing the groups with the greatest relative contribution with respect to the global rate of caesarean sections and observing their variation over time (Figure 1). The groups that reduced their relative contribution to the overall CS rate were Group 1 (nulliparous and spontaneous birth), Group 4 (multiparous and induction), and Group 8 (multiple pregnancies). Group 1 went from 22.1% in 2010 to 16.2% in 2021; similarly, it can be seen how the rate of caesarean sections has decreased (2.3%) during the said period. Group 4 reduced its relative contribution from 13.8% in 2010 to 11.7% in 2021, accompanied by a decrease in the CS rate from 4.5% during that period, and Group 8 from 5.3% in 2010 to 2.8% in 2021, with a reduction in its relative contribution of CS rate of 7.5%. 

Group 2 (nulliparous and induced labour) raised its relative contribution from 23.1% to 39.7%, with the rate of caesarean sections increasing by 6.7%. Group 5 (BVAC) increased its relative contribution to the overall CS rate by 2.7%, and in the time series had an upward behaviour in terms of CS rate, reaching 100% in 2021. Groups 6 (nulliparous and breech), 7 (multiparous and breech), and 9 (abnormal presentation) remained constant. Group 10 (preterm) underwent a 0.6% increase in its relative contribution, although the CS rate increased by 9.1%.

Induction was noted in 33.0% of births. Table 5 sets out the distribution of the indications of induction according to the mode of birth (vaginal or CS). The relative contribution to the overall rate of caesarean sections of Groups 2 and 4 together accounts for 41.2% of the overall rate, and the main indications of induction were premature rupture of membranes (26.1%) and post-term pregnancy (16.3%).

We were interested in analysing the relationship between the onset of labour (spontaneous/induced) and the mode of birth (vaginal/CS). We observed, in Table 6, that the CS rate was higher when labour was induced, and the differences are statistically significant (22.7% vs. 11.3%; *p* < 0.001). There is twice the risk of ending in CS after labour is induced compared to a spontaneous onset delivery (*p <* 0.001).

Table 7 shows the differences between the indications for performing CS between intrapartum and elective caesarean sections. The main indications of intrapartum CS were suspected fetal distress (29.3%), failed induction (22.2%), and obstructed labour (17.9%), while non-cephalic presentation (53.8%) was the most frequent for elective CS.

Regarding the relationship between the main indications of CS and the mode of birth (Table 8), we observed how failed induction has increased in the time series, rising from 25.6% (2010–2013) to 40.1% (2018–2021), with these differences being statistically significant (*p* < 0.001).

## 4. Discussion

This study includes the analysis of the indications of induction of labour and CS of a 12-year time series in a university hospital in eastern Spain, where a total sample of 20,578 births has previously been classified using the RTGCS. An increase in the number of caesarean sections [14,15] has been observed in the groups in which induction was performed.

Although the analysis using the RTGCS constitutes a first step towards investigating differences in CS rates [3,19], the underlying reasons for such differences remain unclear. One of the driving factors behind our adoption of this methodology is our aim to decrease the overall rate of caesarean sections performed in our hospital. In order to establish a target, it is essential to have a clear understanding of the current situation. The upward trend in cesarean sections over the past few decades has highlighted the need for audits that utilize a standardized classification system, such as the ten-group system previously described, which enables comparison of caesarean rates with those of other hospitals. By identifying potential interventions that can help reduce the number of caesarean sections, we hope to make a meaningful contribution towards achieving our goal. Through our secondary analysis of indications of induced delivery and causes of caesarean sections, we have been able to carry out an investigation into the groups that most contribute to the overall rate of caesarean sections. In our study, the prevalence of induction reached 33%, in line with other studies conducted in developed countries ranging from 20% to 40% [20]. The groups with the highest induction rate were 2a (nulliparous induced), 4a (multiparous induced), 5 (vaginal birth after caesarean), 8 (multiple pregnancies), and 10 (preterm birth). Although the distribution of these groups have not increased in all cases in terms of the relative contribution of caesarean sections, the increase in Groups 2a, 5 and 10 stands out, with Group 5 being especially relevant, compared to the historical trend and coinciding with the results reported by the Euro–Peristat Project [3]. The caesarean section rate in Group 5 should not be higher than 60% [1], and we should point out, as in our case, that this may be due to the fact that this group contains a higher number of women with two or more previous caesarean sections.

According to a systematic review and meta-analysis published in the Cochrane Database of Systematic Reviews [21], induction of labour was not associated with an increased risk of caesarean delivery. On the other hand, other studies concluded that induction of labour increased the risk of CS, particularly in nulliparous women that have been classified by the RTGCS as Group 2a, in line with our results [14,15]. In consonance with different authors [6,22], our results show that there is twice the risk of labour ending in a CS when an induction is performed, compared to a spontaneous onset of labour. In addition, we observe that premature membrane rupture and failed induction were, respectively, the most frequent causes of induction and CS, as observed in other studies [12,13,14,23]. In order to reduce the overuse of caesarean sections in this group, efforts should be made to decrease both inductions and elective cesarean sections. Elective cesarean sections should be presented in a clinical session and each specific case should be evaluated, as well as on-demand cesarean sections, which pose a challenge that we must frequently confront. It is important to reflect on whether a caesarean birth prevents an adverse perinatal outcome or, conversely, an unnecessary indication is given in the light of a perceived risk [16]. There is no doubt that in recent years in developed countries, the induction of childbirth has doubled or tripled, while the rate of caesarean sections has been increasing alarmingly [17,24]. This high rate of induction and of CS has failed to reduce adverse perinatal outcomes, and we believe we should be concerned about the short- and long-term effects on maternal morbidity and mortality.

The mean CS rate of our study was 19% (4.5% elective CS rate), which is lower than the average for Spain of around 25% [3]. The main indications of elective CS include non-cephalic presentations (fetal criterion that allows vaginal birth in breech presentation: frank breech or complete breech presentation, estimated fetal weight between 1,500 and 4,000 g, and cephalic attitude in flexion or indifferent, assessed by ultrasound), placenta previa, vasa previa, fetal macrosomia (estimated fetal weight greater than or equal to 4,500 g in diabetic pregnant women and greater than or equal to 5,000 g in non-diabetic pregnant women), presence of uterine scarring, secondary to iterative caesarean, fetal pathology that advises against vaginal birth, and maternal pathology that advises against vaginal birth [25]. In our case, we observed that non-cephalic presentation was the most frequent indication (53.2%), where Groups 6 (nulliparous singleton breeches) and 7 (multiparous singleton breeches) can be included, mainly, together with Groups 8 (multiple pregnancies) and 9 (abnormal lies). Groups 6 and 7 have also shown a greater tendency in CS rates of European countries, in accordance with our results [3].

Furthermore, considering that induction of labour (IOL) is a medical intervention that may affect women’s birth options and their experience of the birth process, this should only be recommended when there are clear indications that continuing with pregnancy poses a greater risk to the mother or baby than the risk of inducing labour [11]. Therefore, in line with previous studies [26,27], the large fraction of women in our study who received IOLs that were not clinically indicated (9.3% “favourable cervix”) or for which the indication for IOL was not documented (5.4%) is of great concern. Having a favourable cervix is not a clinical indication for an induction of labour sustained by the actual scientific evidence [11]. In addition, the indication for IOL must always be documented, and discussion should include reason for induction, method of induction, and risks, including failure to achieve labour and possible increased risk of caesarean section [28]. The appropriate use of IOL has an impact on the resources for performing inductions and the overall CS rate [29].

On the other hand, our results allow us to observe how failed induction [14,15], suspected foetal distress, and CPD are the three main indications of CS during birth.

Addressing the non-medical reasons that drive CS, therefore, is key to reducing its inappropriate use [30]. There is a need to address pre-labour interventions, including IOL [15,31]. In addition, in accordance with the existing literature, factors associated with higher rates of vaginal births may include firm policies on CS due to maternal request, cultural or social pressure, differences in the legal framework for medical litigation, and strategies favouring midwifery-led continuity models of care [32,33,34].

Despite the findings presented in this study, it had several limitations. Firstly, it is a retrospective study, reviewing electronic medical records, and there may be some coding errors inherent in this type of study. Any errors that were detected were analysed by the research team and recoded to reflect the data on the obstetric process collected in the medical record. Secondly, this is a local study, and the results cannot be generalised to the population in the rest of Spain. It is very possible that the hospitals where this systematic approach is applied will obtain different results depending on the degree of obstetric complexity they treat, and that the rates of the highest risk subgroups, such as Groups 6, 7, 8, and 10, will be affected. On the other hand, and with our study objective in mind, subgroups such as Groups 1 to 4 should not be affected by this limitation, which underpins the few publications on this topic.

The strengths of the study lie in its large sample size and the rigorous review of the causes of induction and CS by an experienced research team. The quality of reported data is evident since the methodology that must be implemented when classifying cases was evaluated by the research team. Data quality is confirmed as the research team followed the RTGCS directives to report the data, and the total number of CS and overall number of births reported in this study is identical to those reported by the hospital. In addition, as stated by the RTGCS data quality reporting guidelines, the number of singleton pregnancies in a transverse position (group 9) should be less than 1%, as reported in this study.

## 5. Conclusions

The RTGCS allows comparisons to be made in order to determine the groups in which deviations from the mean occur. Nulliparous pregnant women who underwent induction produced the highest relative contribution of the overall CS rate. The analysis of the causes of induction and the reasons for which caesarean sections are indicated enable the establishment of a starting point to implement changes in clinical protocols with the aim of reducing the number of caesarean sections and maternal morbidity and mortality in future pregnancies. These interventions should be used judiciously, taking into account the specific circumstances of each pregnancy and the potential risks and benefits of each intervention. Considering that the group of primiparous women with induced labour provides the largest relative contribution to the overall rate of caesarean sections in our sample, and that there is also a percentage of IOLs lacking justification, it would be reasonable to think that by reviewing the indications prior to their application, the rate of caesarean sections in that group could be reduced.

## Figures and Tables

**Figure 1 healthcare-11-01521-f001:**
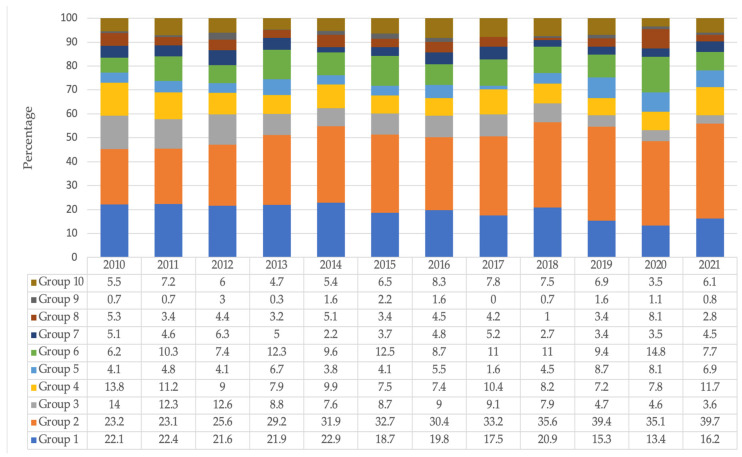
Relative contribution (%) by the group to the overall C-section rate from 2010 to 2021 at HULR 2010–2021 (*N* = 20,578).

**Table 1 healthcare-11-01521-t001:** Group description of Robson’s classification system.

Group	Description
1	Nulliparous, singleton cephalic, ≥37 weeks, spontaneous labour.
2a	Nulliparous, singleton cephalic, ≥37 weeks, induced labour.
2b	Nulliparous, singleton cephalic, ≥37 weeks, or caesarean delivery before labour.
3	Multiparous, singleton cephalic, ≥37 weeks, spontaneous labour.
4a	Multiparous, singleton cephalic, ≥37 weeks, induced labour.
4b	Multiparous, singleton cephalic, ≥37 weeks, caesarean delivery before labour.
5	Previous caesarean delivery, singleton cephalic, ≥37 weeks, spontaneous labour, or induced labour or caesarean delivery before labour (BVAC).
6	All nulliparous singleton breeches, spontaneous labour, or induced labour or caesarean delivery before labour.
7	All multiparous singleton breeches (including previous caesarean delivery), spontaneous labour, or induced labour or caesarean delivery before labour.
8	All multiple pregnancies, spontaneous labour, or induced labour or caesarean delivery before labour.
9	All abnormal singleton lies (including previous caesarean delivery but excluding breech), spontaneous labour, or induced labour or caesarean delivery before labour.
10	All singleton cephalic, ≤36 weeks (including previous caesarean delivery), spontaneous labour, or induced labour or caesarean delivery before labour.

**Table 2 healthcare-11-01521-t002:** Distribution of the type of onset of labour and mode of birth of the sample from 2010 to 2021 at HULR (*N* = 20,578).

		*n*	**%**
**Onset of labour**	Spontaneous	12,770	62.1
	Induced	6776	32.9
	Elective caesarean	916	4.5
	Emergency caesarean	89	0.4
	Emergent caesarean	27	0.1
**Mode of birth**	Vaginal	16,676	81.0
	Emergency caesarean	2986	14.5
	Elective caesarean	916	4.5

**Table 3 healthcare-11-01521-t003:** Distribution of socio-demographic and obstetric variables from 2010 to 2021 at HULR (*N* = 20,578).

		2010–2013		2014–2017		2018–2021		
		*n*	*%*	*n*	*%*	*n*	*%*	*p*-Value *
**Country of origin**	Spain	6897	83.0	5570	81.9	3877	70.9	<0.001
	Foreign	1411	17.0	1235	18.1	1588	29.1	
**Newborn sex**	Male	4290	51.6	3486	51.2	2802	51.3	0.897
	Female	4018	48.4	3319	48.8	2663	48.7	
**Previous CS**	No	8077	97.2	6644	97.6	5313	97.2	0.216
	Yes	231	2.8	161	2.4	152	2.8	
**Number of fetus**	One	8216	98.9	6725	98.8	5402	98.8	0.928
	Two or more	92	1.1	80	1.2	63	1.2	
**Previous pregnancies**	One	4321	55.6	3835	61.0	3121	63.9	<0.001
	Two	3450	44.4	2448	39.0	1766	36.1	
	Three	466	86.8	438	83.9	431	74.6	
	**Four or more**	71	13.2%	84	16.1	147	25.4	
			**Period**		** *n* **	**Mean**	** *SD* **	***p*-value ****
**Maternal age**			2010–2013		8308	30.4	5.5	<0.001
			2014–2017		6805	31.1	5.7	
			2018–2021		5465	31.2	6.1	
**Number of births**			2010–2013		8308	0.6	0.7	<0.001
			2014–2017		6805	0.5	0.7	
			2018–2021		5465	0.6	0.7	
**Birth weight**			2010–2013		8308	3281.1	466.5	0.022
			2014–2017		6805	3280.0	472.5	
			2018–2021		5465	3304.7	484.5	

* Chi-square test; ** Kruskal–Wallis test; HULR: Hospital Universitario de la Ribera; CS: cesarean section.

**Table 4 healthcare-11-01521-t004:** Distribution of births using the Robson classification system from 2010 to 2021 at HULR (*N* = 20,578).

Group	C-Section in the Group	No. of Women in the Group	Group Size	C-Section Rate of the Group	Absolute Contribution by the Group to the Overall C-Section Rate	Relative Contribution by the Group to the Overall C-Section Rate
1	784	6842	33.2%	11.5%	3.8%	20.1%
2	1228	3272	15.9%	37.5%	6.0%	31.5%
3	359	6427	31.2%	5.6%	1.7%	9.2%
4	377	1830	8.9%	20.6%	1.8%	9.7%
5	205	317	1.5%	64.7%	1.0%	5.3%
6	323	402	2.0%	80.3%	1.6%	8.3%
7	172	179	0.9%	96.1%	0.8%	4.4%
8	163	284	1.4%	57.4%	0.8%	4.2%
9	40	47	0.2%	85.1%	0.2%	1.0%
10	251	978	4.8%	25.7%	1.2%	6.4%
	3902	20,578	100%	19.0%	19.0%	100%

**Table 5 healthcare-11-01521-t005:** Distribution of indications of induction and mode of birth from 2010 to 2021 at HULR (*N* = 6776).

	Vaginal (*n* = 5265)		C-Section (*n* = 1511)	
	*n*	*%*	*n*	*%*
Intrauterine fetal death	30	100	0	0
Unmonitored gestation	1	50.0	1	50.0
Indication of induction not documented	362	99.7	1	0.3
Placenta previa	0	0	1	100
Anhydramnios	32	94.1	2	5.9
Poor obstetric history	5	71.4	2	28.6
Reduced fetal movements	1	33.3	2	66.7
Prior CS	9	81.8	2	18.2
Advanced maternal age	2	40.0	3	60.0
Fetal pathology	6	66.7	3	33.3
Favourable cervix	628	99.4	4	0.6
3rd trimester metrorrhagia	33	86.8	5	13.2
Insidious prepartum with poor pain control	366	90.0	10	10.0
Doppler alterations	20	64.5	11	35.5
Twin pregnancy	26	60.5	17	39.5
Suspected macrosomia	70	74.5	24	25.5
Maternal disease	77	74.0	27	26.0
Polyhydramnios	59	64.8	32	35.2
Small for gestational age	113	74.8	38	25.2
Gestational diabetes	90	61.2	57	38.8
Restricted intrauterine growth	189	73.3	69	26.7
Cardiotocographic anomalies	203	68.1	95	31.9
Meconial amniotic fluid	363	76.9	109	23.1
Oligohydramnios	283	70.6	117	29.4
Pre-eclampsia	170	56.7	130	43.3
Post-term pregnancy	763	69.3	344	30.7
Premature rupture of membranes	1364	79.0	405	21.0

**Table 6 healthcare-11-01521-t006:** Relationship between induction and mode of birth from 2010 to 2021 at HULR (*N* = 19,662).

		Vaginal		C-Section					
		*n*	*%*	*n*	*%*	*p*-Value *	OR	CI 95%	*p*-Value **
**Onset of labour**	Spontaneous	11,427	88.8	1448	11.3	<0.001	1		<0.001
	Induced	5249	77.2	1538	22.7		2.3	2.1–2.5	

* Chi-square test; ** Simple logistic regression.

**Table 7 healthcare-11-01521-t007:** Distribution of the indication of caesarean section from 2010 to 2021 at HULR (*N* = 3902).

	Intrapartum C-Section (*n* = 2986)		Elective C-Section (*n* = 916)	
	*n*	*%*	*n*	*%*
Restricted intrauterine growth	4	14.3	24	85.7
Triplets	0	0.0	1	100
Fetal pathology	1	14.3	6	85.7
Prematurity	1	100	0	0
Prior uterine surgery	2	7.4	25	92.6
Uterine rupture	2	100	0	0
Chorioamnionitis	3	100	0	0
Favourable cervix	6	33.3	12	66.7
Maternal pathology	22	33.3	44	66.7
Cord prolapse	16	100	0	0
Suspected macrosomia	17	14.5	117	85.5
Iterative caesarean section	19	12.8	149	87.2
Placenta previa	20	41.7	28	58.3
Poorly controlled preeclampsia	33	60.0	22	40.0
Placental abruption	56	100	0	0
Fetal malposition	59	100	0	0
Non-cephalic presentation	254	36.3	487	63.7
Cephalopelvic disproportion	396	99.7	1	0.3
Obstructed labour	534	100	0	0
Failed induction	664	100	0	0
Suspected fetal distress	877	100	0	0

**Table 8 healthcare-11-01521-t008:** Indication of intrapartum caesarean section and mode of birth in time series from 2010 to 2021 at HULR (*N* = 1949).

		2010–2013		2014–2017		2018–2021		
		*n*	*%*	*n*	*%*	*n*	*%*	*p*-Value *
**Indication of** **C-section**	CPD	200	24.3	100	16.9	96	18.2	<0.001
	Failed induction	200	25.6	250	42.3	214	40.1	
	SFD	413	50.1	241	40.8	223	41.7	

* Chi-square test; CPD: cephalopelvic disproportion; SFD: suspected fetal distress.

## Data Availability

All necessary data are supplied and available in the manuscript; however, the corresponding author will provide the dataset upon request.

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
