# Peer review of "Indications of Induction and Caesarean Sections Performed Using the Robson Classification in a University Hospital in Spain from 2010 to 2021"

_healthcare, 2023, doi:10.3390/healthcare11111521_

Round 1
Reviewer 1 Report
Review of healthcare-2308891
This paper categorizes inductions and caesarian sections in a single hospital in Spain over a course of 12 years. As a description of the situation, it is fine, but there is no real analysis. I am not really sure what the point of this paper is. Do they just want to count cases? Do they want to intervene in the subsets where induction and CS are not medically indicated? For that they might want an analysis using more of the sociodemographic variables. Personally, I would like to see more analysis of the trends over time and their relationships (both the outcomes and the possible predisposing variables, e.g., maternal age and citizenship, parity, multiplicity, breech presentation, ?others). Lines 145-153 are really interesting.
Also, the word “causes” in the title seems unwarranted.
The authors discuss trends over time, but do not link these with possible demographic trends (maternal age, maternal ethnicity (Spanish vs not, at least), and parity) or fetal size (except as this is captured in the RTGCS. It is also possible that there are seasonal and day-of-week effects, which should be distinguishable (at least for rates of induction and CS, though not necessarily for the smaller categories) in a data set of this size, assuming that the dates of delivery (or the dates of start of labour—pick one) are available.
IMPORTANT METHODOLOGICAL ISSUE: Over 12 years, it is likely that some of these deliveries are “repeats” of women. This would mean that the observations are not necessarily independent. If the investigators have any identifier that can be used to match up the same women, they should at least look at the distribution of number of observations per woman (e.g., 1, 2, 3+ would probably do). If the number of repeats is large, some method that takes clustering into account, such as GEE, would be warranted.
The methods section describes multiple logistic regression, but I don’t see any evidence of this. The authors also discuss Bonferroni correction for subgroup analysis. As far as I can see, there are only 3 p-values in this paper, and no real subgroup analyses. It is not clear what analysis produced these p-values. It looks as if this section has been lifted from another paper, probably by some of the same authors. I’m not worried about plagiarism, so much as the indication that the authors didn’t think about what they were writing.
All titles and tables should include the location (Spain, or perhaps more specifically eastern Spain) and the years covered by the data (2010-2021). Tables with p-values should note the analysis done to arrive at the p-value(s). Tables should be internally consistent in the number of decimal places. In general, it does not help readers to report percentages to more than one decimal place.
It also does not help readers to report odds ratios or their CIs to 3 decimal places (especially when the OR is only reported to 2).
Specific comments
Line comment
. 35-36 Obviously, some of the RTGCS should have higher fractions of CS. It is not clear what the authors are trying to say. In some of these groups, it is possible that the optimal rate of CS is 100%, for example. Whatever the “optimal rate” is, it depends on the presentation and medical status of the birthing women.
. 82 This line says that there were about 1300 births per year, but the total number of births in the study is 20578, or about 1700 per year. The 1300 number may be a carry-over from some previous publication or report.
. 118-119 What scale was used for the 10th and 5th percentiles?
. 130 If the authors end up doing any multiple logistic regression, they should phrase it as I have here (not “logistic model adjustment”).
. 136 I gather this is the number after deleting the ineligible (BW < 500g, GA at delivery <= 22 weeks).
. Table 1 Perhaps “Delivery” is better than “End of labour.”
. What does “emerging caesarian” mean?
. 140 “The CS rate,” not ”The mean CS rate.”
. At onset of labour, all caesarians are 5% of the total (1032/20578), and elective CS are 4.5%. At delivery 19% are CS, of which 77% are non-elective.
. 149 What does breastfeeding after delivery have to do with the general subject matter of this paper (other than that the authors seem to have captured it in their data)?
. Table 2 All the percentages end in 0 (in the second decimal place). Clearly these are rounded or just have 0s added to a previously written 1 decimal place. Get rid of the extra 0s. While it is true that groups 1-4 contribute most of the c-sections, they are very large and have low rates. Groups 5-9 are small, but have very high rates (which are probably justified). This issue of extra zeroes seems to be true in other tables as well.
. Fig 1 Is the sum of group 1 and group 2 (apparently “the same” except for the type of delivery) stable over time? Almost all the increase over time comes from this group (>= 37 weeks, cephalic presentation). Please explain in a footnote that the numbers on the graph (if they are even worth keeping there) are the percent of the group with c-sections.
. Table 3 (title) Please change “end of birth” to “type of delivery”
. 183-187 Your comment is correct, but the situation of people who have induced labor is different from that of those who don’t. It would be instructive to look at the combined groups 1, 2a, 2b and determine whether the c-section rate is higher in those not induced. This assumes that a woman with induced labor who has a c-section is categorized as 2b. How many “failed inductions” in Table 5 were from groups 1, 2a, 2b?
. Table 4 This table seems to come from logistic regression. What is the first p-value from? Some sort of contingency table analysis (which is not mentioned in the methods)? Were the 6-8th columns produced by multiple logistic regression? If so, please specify in a footnote what variables, if any, were adjusted for.
. Table 5 Over half (not just the most frequent cause) of the elective c-sections were for non-cephalic presentation.
. This table might be more interesting if there were a third column (no c-section), so that the comparison is not merely which type of c-section was done, but whether one was done at all.
. 196-198 To what extent is the increase of the fraction of c-sections due to failed induction related to an increase in the number of inductions? Actually, failed inductions are only slightly up. The change in attribution of the intrapartum c-sections to failed induction resulted from the great decreases (by about 50%) for the other two reasons shown. Does this represent changes in criteria for (or more precise determination of) these two reasons? That would certainly be an interesting finding in this paper. Does the apparent decrease in births over time in this hospital represent a more general trend (see regional or national statistics) or a change of “market share” of this hospital?
. Table 6 The numbers for CPD and FI in this table are inconsistent with those in Table 5.
. Please do an explicit trend test (and describe it in the Methods section).
. 221 “Coinciding with” should probably be “Agreeing with” or “In consonance with.” This is just an issue of idiomatic English.
. 251-253 Reasons for induction: state which lines in Table 3 are “no clear clinical justification.” I suspect there is some disagreement about some of these categories.
. Previous c-section 11/6776. Can the authors read their own tables? Not described 363/6776 (a bit over 5%). “favourable cervix” 632/6776 (a bit under 10%, not .6% as written). PLEASE CHECK ALL NUMBERS IN THIS PAPER.
Author Response
Response to Reviewer 1 Comments
Thank you for your positive and constructive feedback on our manuscript.
We have considered all your comments and suggestions and also the comments made by the others reviewers attempting to improve/refine the original manuscript.
Below you will find a point-by-point response to your comments (in red).
Point 1: This paper categorizes inductions and caesarian sections in a single hospital in Spain over a course of 12 years. As a description of the situation, it is fine, but there is no real analysis. I am not really sure what the point of this paper is. Do they just want to count cases? Do they want to intervene in the subsets where induction and CS are not medically indicated? For that they might want an analysis using more of the sociodemographic variables. Personally, I would like to see more analysis of the trends over time and their relationships (both the outcomes and the possible predisposing variables, e.g., maternal age and citizenship, parity, multiplicity, breech presentation, ?others). Lines 145-153 are really interesting.
Response 1: Thank you for your comment. The aims of the study were to examine all births attended in the period 2010-2021 using Robson's 10-step classification system (RTGCS), assess their trend descriptively (lines 22-23), describe the indications for induction of labour and caesarean sections performed and examine the association between induction of labour and caesarean section.
The follow-up years were divided into 3 periods of time (2010-2013, 2014-2017, and 2018-2021) so these could be compared. Analyses of the sociodemographic variables (country of origin and maternal age) and obstetric variables that do not require to be included in the RTGCS (sex of the new born, birth weight, indication for induction and cause of indication for caesarean section) were performed for the total sample and for each one of the 10 Robson groups separately. The resulting analysis generates 20 different tables, 10 for the quantitative variables using the Kruskall-Wallis test, and 10 with the categorical variables, using the chi-square test. The research team agreed to describe the analysis in a general way and not by subgroups because the Robson classification already groups women according to 5 obstetric criteria (parity, previous caesarean sections, foetal presentation, gestational age, and onset of labour). The analysis of the obstetric variables for each of the groups should not be considered due to the correlation between the obstetric variables and those that define the Robson groups. For example, if we analyse parity, in group 1 there will be no statistical significance because all the women in that group are primiparous, and similarly, the same will occur with the number of foetuses, foetal presentation, and previous caesarean section. Thus, the relationships studied in general were described in text form and not in a table.
In the same way, an analysis of the subgroups was carried out for each of the periods for the 3 most frequent causes of urgent caesarean section (pelvis-foetal disproportion, induction failure and suspicion of loss of foetal well-being), for the 3 established periods, although only the relationship between the causes of caesarean section and the time series were described (new Table 6).
In accordance with this comment, we have included a new table 2 (pages 4-5), which describes the sociodemographic and obstetric characteristics of the time series.
Point 2: Also, the word “causes” in the title seems unwarranted.
Response 2: Thank you for your comment. According to your comment, we have now changed the word “causes” to “indications” throughout our manuscript.
Point 3: The authors discuss trends over time, but do not link these with possible demographic trends (maternal age, maternal ethnicity (Spanish vs not, at least), and parity) or fetal size (except as this is captured in the RTGCS. It is also possible that there are seasonal and day-of-week effects, which should be distinguishable (at least for rates of induction and CS, though not necessarily for the smaller categories) in a data set of this size, assuming that the dates of delivery (or the dates of start of labour—pick one) are available.
Response 3: Thanks for your reflection. As for analysing trends in terms of obstetric variables, as we have already commented in Response 1, in this case parity is one of the defining characteristics of the Robson's groups. Although it would be very interesting to explore the trends to see if there is seasonality, we do not have this data, but births grouped by year, since that was not our objective of the study.
Point 4: IMPORTANT METHODOLOGICAL ISSUE: Over 12 years, it is likely that some of these deliveries are “repeats” of women. This would mean that the observations are not necessarily independent. If the investigators have any identifier that can be used to match up the same women, they should at least look at the distribution of number of observations per woman (e.g., 1, 2, 3+ would probably do). If the number of repeats is large, some method that takes clustering into account, such as GEE, would be warranted.
Response 4: The frequency of repeated women was 11.5% (2366/20512). Carrying out a multivariate analysis with Generalized Estimation Equations, which allows data to be modelled with repeated measures, does not fall within our objectives because this study intends to carry out a descriptive approximation of temporal trends. We have included in the objective of the study that it will be carried out from a descriptive approach (lines 22-23, and 79).
Point 5: The methods section describes multiple logistic regression, but I don’t see any evidence of this. The authors also discuss Bonferroni correction for subgroup analysis. As far as I can see, there are only 3 p-values in this paper, and no real subgroup analyses. It is not clear what analysis produced these p-values. It looks as if this section has been lifted from another paper, probably by some of the same authors. I’m not worried about plagiarism, so much as the indication that the authors didn’t think about what they were writing.
Response 5: Thank you for your comment. All the analyses described were performed, not being any copy of another manuscript. The logistic regression can be seen in Table 5 (page 8). The statistical tests used to obtain the p-value (chi-square test and simple logistic regression) are detailed. In addition, the OR and the 95% confidence interval are provided. Regarding the analysis of the groups for each of the sociodemographic and obstetric variables for the time series, these were performed although they were neither described nor exposed.
In order to include the data to of the analysis described, a new paragraph has been rewritten in lines 185-195 and Supplementary Table 1 has been added.
Point 6: All titles and tables should include the location (Spain, or perhaps more specifically eastern Spain) and the years covered by the data (2010-2021). Tables with p-values should note the analysis done to arrive at the p-value(s). Tables should be internally consistent in the number of decimal places. In general, it does not help readers to report percentages to more than one decimal place.
It also does not help readers to report odds ratios or their CIs to 3 decimal places (especially when the OR is only reported to 2).
Response 6: Thank you very much for the comment. We understand that including Spain or Eastern Spain would not be representative, so we opted to include HULR. We find it appropriate to include the date in each of the tables.
Regarding the p-value of the tables, it has been corrected, as has the number of decimal places, as suggested by the reviewer.
Specific comments
Point 7: Lines 35-36 Obviously, some of the RTGCS should have higher fractions of CS. It is not clear what the authors are trying to say. In some of these groups, it is possible that the optimal rate of CS is 100%, for example. Whatever the “optimal rate” is, it depends on the presentation and medical status of the birthing women.
Response 7: Thank you for your comment. Optimal caesarean section rates are determined by Robson's manual (https://www.who.int/publications/i/item/9789241513197). We think that the expression “optimal” is adequate.
Point 8: Line 82 This line says that there were about 1300 births per year, but the total number of births in the study is 20578, or about 1700 per year. The 1300 number may be a carry-over from some previous publication or report.
Response 8: Thank you for your comment. With respect to the total number of births, we have described the current number of births per year rather than the average of the 12 years. We believe that this represents better the current state of care since in Spain the birth rate has been gradually decreasing each year since 2008-2009.
Point 9: Lines 118-119 What scale was used for the 10th and 5th percentiles?
Response 9: Thanks for the comment. Reference number 18 has been added.
Point 10: Line 130 If the authors end up doing any multiple logistic regression, they should phrase it as I have here (not “logistic model adjustment”).
Response 10: Thanks for the comment. It was only used the simple logistic regression to determine the relationship between the onset of labour and the mode of birth without using any variable as adjustment. We have specify in line 142-143 this comment.
Point 11: Line 136 I gather this is the number after deleting the ineligible (BW < 500g, GA at delivery <= 22 weeks).
Response 11: Thanks for the comment. Indeed, it is the total sample analysed.
Point 12: Table 1 Perhaps “Delivery” is better than “End of labour.”
Response 12: Gracias por el comentario. Amended as mode of birth in all tables and manuscript sections.
Point 13: What does “emerging caesarian” mean?
Response 13: We do apologise as this was a language typo. It is “emergent” caesarean section (CS) rather than “emerging” CS. This has now been amended in the manuscript. Caesarean sections can be classified: emergency CS, emergent CS and elective CS. In the case of emergency CSs, the safe time between diagnosis and the start of the caesarean section should not exceed 30 minutes (international standard). In the case of an emergent caesarean section, the time between diagnosis and the start of the intervention should not exceed 15 minutes.
Point 14: Line 140 “The CS rate,” not ”The mean CS rate.”
Response 14: Thank you. This has now been amended in line 153.
Point 15: At onset of labour, all caesarians are 5% of the total (1032/20578), and elective CS are 4.5%. At delivery 19% are CS, of which 77% are non-elective.
Response 15: The rate of caesarean sections must include the three types previously explained (emergency, emergent and elective). We understand that as the results are presented in Table 1, the distribution of caesarean sections is clear, since calculating (4.5/19)-100 to obtain 77% is more complicated to understand. We have decided to include the term “respectively” on line 144 to clarify the description.
Point 16: Line 149 What does breastfeeding after delivery have to do with the general subject matter of this paper (other than that the authors seem to have captured it in their data)?
Response 16: Thank you for your feedback. We agree with the comment. We have decided to eliminate the breastfeeding variable.
Point 17: Table 2 All the percentages end in 0 (in the second decimal place). Clearly these are rounded or just have 0s added to a previously written 1 decimal place. Get rid of the extra 0s. While it is true that groups 1-4 contribute most of the c-sections, they are very large and have low rates. Groups 5-9 are small, but have very high rates (which are probably justified). This issue of extra zeroes seems to be true in other tables as well.
Response 17: Thank you for your highlight. The second decimal has been amended in all presented tables.
Point 18: Fig 1 Is the sum of group 1 and group 2 (apparently “the same” except for the type of delivery) stable over time? Almost all the increase over time comes from this group (>= 37 weeks, cephalic presentation). Please explain in a footnote that the numbers on the graph (if they are even worth keeping there) are the percent of the group with c-sections.
Response 18: Thank you for your input. In order to clarify the Figure’s 1 title, we have rewritten it as “Relative contribution (%) by the group to the overall c-section rate from 2010 to 2021 at HULR 2010-2021 (N=20,578).”
Point 19: Table 3 (title) Please change “end of birth” to “type of delivery”
Response 19: Thank you for your comment. This has now been amended throughout the manuscript as mode of birth as recommended by RCM, ICM, RCOG.
Point 20: Lines 183-187 Your comment is correct, but the situation of people who have induced labor is different from that of those who don’t. It would be instructive to look at the combined groups 1, 2a, 2b and determine whether the c-section rate is higher in those not induced. This assumes that a woman with induced labor who has a c-section is categorized as 2b. How many “failed inductions” in Table 5 were from groups 1, 2a, 2b?
Response 20: Thanks for your reflection. We do apologise as we do not understand the comment because the women in group 1 have not been induced, these are births with a spontaneous onset of labour that can end in caesarean section or vaginal delivery. As explained in the discussion (lines 262-267), the groups in which an induction can be indicated are groups 2a, 4a, 5, 8 and 10. We were interested in analysing the total number of inductions and not a sub-analysis of the groups.
Point 21: Table 4 This table seems to come from logistic regression. What is the first p-value from? Some sort of contingency table analysis (which is not mentioned in the methods)? Were the 6-8th columns produced by multiple logistic regression? If so, please specify in a footnote what variables, if any, were adjusted for.
Response 21: Thank you for your highlight. Now, table 5 (page 8) has been amended. Simple logistic regression and chi-square test has been used, as indicated in Response 5.
Point 22: Table 5 Over half (not just the most frequent cause) of the elective c-sections were for non-cephalic presentation.
Response 22: Indeed, 53.8% of caesarean sections were performed for this reason. As an example, in this case twin pregnancies can be found, whose first twin is breech, or transverse, or both. Please consider that this table (now table 6) describes the indications for caesarean sections performed in all groups.
Point 23: This table might be more interesting if there were a third column (no c-section), so that the comparison is not merely which type of c-section was done, but whether one was done at all.
Response 23: Thank you for your comment. We are not sure we understand what the reviewer means. We cannot include a third column for non-caesarean section because it would be empty, since it would not meet any of the indications for caesarean section because they would be vaginal deliveries. Please, in the case amendments require to be done in the manuscript, we would appreciate if this comment were clarified.
Point 24: Lines 196-198 To what extent is the increase of the fraction of c-sections due to failed induction related to an increase in the number of inductions? Actually, failed inductions are only slightly up. The change in attribution of the intrapartum c-sections to failed induction resulted from the great decreases (by about 50%) for the other two reasons shown. Does this represent changes in criteria for (or more precise determination of) these two reasons? That would certainly be an interesting finding in this paper. Does the apparent decrease in births over time in this hospital represent a more general trend (see regional or national statistics) or a change of “market share” of this hospital?
Response 24: Thank you for your comments. In relation to what extent is the in the increase of the fraction of c-sections due to failed induction related to an increase in the number of inductions, we agree that this a very interesting question. Unfortunately, it is not possible to provide a straightforward answer. The extent would really depend on several factors such as women’s parity, the indication of the IOL, the gestational age when the IOL is performed, the alternative provided to IOL (expectant management or CS), etc. This is something that would be really interesting to consider in future research with a different analysis approach.
Bhide, A. (2021), Induction of labor and cesarean section. Acta Obstet Gynecol Scand, 100: 187-188. https://doi.org/10.1111/aogs.14068
Regarding the decrease in births over time in this hospital, this is correlated to the national decrease of births. Please, see the trends of the total number of births in Spain since 2010-2021 from the National Statistical Institute of Spain: https://www.ine.es/jaxiT3/Datos.htm?t=6506
Point 25: Table 6 The numbers for CPD and FI in this table are inconsistent with those in Table 5.
Response 25: Thank you for the highlight. Indeed, these are two typographical errors. The percentages were correct. This has now been amended, now in table 7.
Point 26: Please do an explicit trend test (and describe it in the Methods section).
Response 26: Thank you for the comment. As explained in Response 4, we have included in the objective of the study that it will be carried out from a descriptive approach (lines 22-23, and 79).
Point 27: Line 221 “Coinciding with” should probably be “Agreeing with” or “In consonance with.” This is just an issue of idiomatic English.
Response 27: Thank you for your correction. Amended in line 274.
Point 28: Lines 251-253 Reasons for induction: state which lines in Table 3 are “no clear clinical justification.” I suspect there is some disagreement about some of these categories.
Response 28: Thank you for your comment. As stated, now in Table 4, 363 (5.4%) of women that underwent an induction of labour (IOL) in our study had no clear clinical justification documented. We have now amended this line 311 (previous line 251).
In addition, as mentioned in lines 310-311, we also observed that there were women who received an induction of labour with the indication of having “a favourable cervix”. Having a favourable cervix should only be considered as a clinical observation to decide the method of induction and not as an indication of induction of labour by itself.
World Health Organization. WHO recommendations on induction of labour, at or beyond term; Geneva, 2022; ISBN 9789240052796.
Point 29: Previous c-section 11/6776. Can the authors read their own tables? Not described 363/6776 (a bit over 5%). “favourable cervix” 632/6776 (a bit under 10%, not .6% as written). PLEASE CHECK ALL NUMBERS IN THIS PAPER.
Response 29: Thank you for your highlight. We have now amended this.

Reviewer 2 Report
Dear Authors,
Thank you for this wonderful piece of a retrospective study. Kindly find the attached pdf with my comments. I would like you to clarify the knowledge gap in the introduction and revisit the discussion section. Lastly, read through the article carefully and correct the grammatical errors and some glaring omissions which perhaps could change the meaning of some sentences. The editor will provide additional comments.
Thank you

Author Response
Response to Reviewer 2 Comments
Thank you for your very positive and constructive feedback on our manuscript.
We have considered all your comments and suggestions (and also the comments made by the other reviewer) attempting to improve/refine the original manuscript.
Below you will find a point-by-point response to your comments (in red).
Dear Authors,
Thank you for this wonderful piece of a retrospective study. Kindly find the attached pdf with my comments. I would like you to clarify the knowledge gap in the introduction and revisit the discussion section. Lastly, read through the article carefully and correct the grammatical errors and some glaring omissions which perhaps could change the meaning of some sentences. The editor will provide additional comments.
Thank you
Point 1: To have a grounded introduction in the abstract, Kindly consider revising line 19-20 to include..." assessing, monitoring and comparing caesarean section rates both within healthcare facilities and between them." as the role of RTGCS.
Response 1: Thank you for your help. This has now been amended in lines 20-21.
Pont 2: Introduction second line, include the preposition 'to' after due....
Response 2: Thank you for your comment. This has now been amended in line 46.
Point 3: Introduction: In line 53 ...Just say In 2022, rather than recently.
Response 3: Thank you for your comment. The year has been included in line 57.
Point 4: Line 61....Typo...a preposition missing
Response 4: Thank you for your comment. Unfortunately, we have not been able to find where is the preposition missing.
Point 5: Line 62 ....tense..should read "increased from two.......
Response 5: Thank you for your comment. Amended in line 66.
Point 6: Line 64 can be revised from passive to active sentence:
I suggest you can revise it to read: "Previous studies (12,13), have associated the induction labour with negative outcomes such as.........."
Response 6: Thank you for your comment. The paragraph has been improved with your suggestion in lines 68-69.
Point 7: Line 67 ..active sentence...According to studies by.....(CITE AUTHORS HERE i.e. 14,15)
Response 7: Thank you for your comment. Amended in lines 71-72
Point 8: In the last paragraph of introduction, line 73-76, there is need for authors to show in the background what is so interesting in La Ribera University Hospital (Spain) as compared to other hospitals in Spain that ignites need for this analysis.
Response 8: Thank you for your input. It is recommended by the WHO that the Robson classification system is adopted in all settings in order to classify, monitor and compare CS rates in a standardized, reliable, consistent and action-oriented manner with the aim of understanding the drivers and contributors of this trend across different settings and populations and identify areas where interventions may be needed to reduce CS rates as specified in lines 51-54. Therefore, our hospital merely abides by the recommendations of the World Health Organization, with the aim of reducing the cesarean section rate in our facility (reference 7):
Betrán, A.P.; Temmerman, M.; Kingdon, C.; Mohiddin, A.; Opiyo, N.; Torloni, M.R.; Zhang, J.; Musana, O.; Wanyonyi, S.Z.; Gülmezoglu, A.M.; et al. Interventions to reduce unnecessary caesarean sections in healthy women and babies. Lancet 2018.
Point 9: Data were obtained ....by who?? consider writing active sentences.
Response 9: Thank you for your comment. Amended in line 87. We have also considered specifying how we handle the data in order to tabulate the sample in lines 118-119 and 129-132.
Point 10: line 85-85 authors to revise for clarity.......maybe just say we obtained ethical clearance from........name the body.
Response 10: Thank you for your comment. Amended in line 92.
Point 11: Now explain the implications of this. Then compare and contrast with previous studies (discuss)
Response 10: Thank you for your input. A new paragraph has been written in lines 307-317.

Reviewer 3 Report
In my opinion, the study will contribute to the literature. The retrospective study examines a 12 years period in an university hospital datas.
Author Response
Response to Reviewer 3 Comments
Thank you for your very positive and constructive feedback on our manuscript.
In my opinion, the study will contribute to the literature. The retrospective study examines a 12 years period in an university hospital datas.
Response 1: Thank for your comment.

Reviewer 4 Report
This is a well written study which conducted an analysis to assess levels and trends of births by CS in La Ribera University Hospital (Spain) between 2010-2021 using the Robson classification. Additionally, they also described the indications for inducing labor and the causes of cesarean sections performed, and examined the association between induced labor and CS birth. Overall the paper discusses the results clearly along with conclusions and limitations.
I have a few minor comments to offer for the paper:
Figure 1 could probably be visualized differently or improved. It has many groups with overlapping lines and colors, which could potentially make it a bit difficult for readership.
Line 255 probably has a white line included by accident
some detailed comments:
1.What is the main question addressed by the research?
The aims of the present study were to conduct an analysis to assess levels and trends of births by CS in La Ribera University Hospital (Spain) between 2010-2021 using the Robson classification, describe the indications for inducing labor and the causes of cesarean sections performed, and examine the association between induced labor and CS birth.
2. Do you consider the topic original or relevant in the field? Does it address a specific gap in the field?
The topic is of significance. It describes the indications for inducing labor and the causes of cesarean sections performed, and examined the association between induced labor and CS birth. It addresses the lack of consensus on the appropriate CS rate, the associated short- and long-term maternal and neonatal risks and costs, and the inequity in access.
3. What does it add to the subject area compared with other published material?
It addresses the lack of consensus on the appropriate CS rate, the associated short- and long-term maternal and neonatal risks and costs, and the inequity in access. Therefore, the aims of the present study were to conduct an analysis to assess levels and trends of births by CS in La Ribera University Hospital (Spain) between 2010-2021 using the Robson classification, describe the indications for inducing labor and the causes of cesarean sections performed, and examine the association between induced labor and CS birth. Prior studies have generally not used the Robson classification.
4. What specific improvements should the authors consider regarding the methodology? What further controls should be considered?
The study will be more powerful if the results can be compared using assessments performed by previous studies.
5. Are the conclusions consistent with the evidence and arguments presented and do they address the main question posed?
yes it address the main questions
6. Are the references appropriate?
yes references are appropriate
Author Response
Response to Reviewer 4 Comments Round 1
Thank you for your very positive and constructive feedback on our manuscript.
We have considered all your comments and suggestions (and also the comments made by the other reviewer) attempting to improve/refine the original manuscript.
Below you will find a point-by-point response to your comments (in red).
Point 1: This is a well written study which conducted an analysis to assess levels and trends of births by CS in La Ribera University Hospital (Spain) between 2010-2021 using the Robson classification. Additionally, they also described the indications for inducing labor and the causes of cesarean sections performed, and examined the association between induced labor and CS birth. Overall the paper discusses the results clearly along with conclusions and limitations.
Response 1: Thank you for your support. We would like to clarify that your review was provided to us after the first round had already been completed, and that you reviewed the original manuscript rather than the round 1 revision. We are currently in round 2.
I have a few minor comments to offer for the paper:
Point 2: Figure 1 could probably be visualized differently or improved. It has many groups with overlapping lines and colors, which could potentially make it a bit difficult for readership.
Response 2: Thank you for your comment. A new figure with % grouped bars has been created for improved visualization in page 6 (round 2).
Point 3: Line 255 probably has a white line included by accident
Response 3: Thank you for your comment. Indeed, that line was deleted.
Point 4: some detailed comments:
1.What is the main question addressed by the research?
The aims of the present study were to conduct an analysis to assess levels and trends of births by CS in La Ribera University Hospital (Spain) between 2010-2021 using the Robson classification, describe the indications for inducing laborand the causes of cesarean sections performed, and examine the association between induced labor and CS birth.
Do you consider the topic original or relevant in the field? Does it address a specific gap in the field?
The topic is of significance. It describes the indications for inducing labor and the causes of cesarean sections performed, and examined the association between induced labor and CS birth. It addresses the lack of consensus on the appropriate CS rate, the associated short- and long-term maternal and neonatal risks and costs, and the inequity in access.
What does it add to the subject area compared with other published material?
It addresses the lack of consensus on the appropriate CS rate, the associated short- and long-term maternal and neonatal risks and costs, and the inequity in access. Therefore, the aims of the present study were to conduct an analysis to assess levels and trends of births by CS in La Ribera University Hospital (Spain) between 2010-2021 using the Robson classification, describe the indications for inducing labor and the causes of cesarean sections performed, and examine the association between induced labor and CS birth. Prior studies have generally not used the Robson classification.
Response 4: Thank you for your positive comments on our manuscript. We truly appreciate your feedback.
Point 5:
- What specific improvements should the authors consider regarding the methodology? What further controls should be considered?
The study will be more powerful if the results can be compared using assessments performed by previous studies.
Response 5: Thank you for your comment. The manuscript has been improved by expanding the discussion and conducting new analyses requested by other reviewers. Please, consider the round 2 reviewed manuscript.
Point 6:
- Are the conclusions consistent with the evidence and arguments presented and do they address the main question posed?
yes it address the main questions
Are the references appropriate?
yes references are appropriate
Response 6: Thank you very much for taking the time to review our work and providing us with your valuable feedback. Your positive comments are greatly appreciated and serve as encouragement to continue our research endeavors. We are thrilled to hear that our research has resonated with you, and we hope that it can make a meaningful contribution to the field. Thank you again for your time and consideration.

Reviewer 5 Report
Review report for autors:
A brief summary:
The aims of this study were to analys assess levels and trends of birth by Cesarean sections in La Ribera University Hospital , Spain during 11 - years period, using the Robson classification, describe the indications for induction of labour and the causes of caesarean sections performed, also they examine the association between induction of labour and birth by Cesaren section.
The increase in the number and percentage of Caesarean sections is a major public health problem, with a trend of growth over the years in all countries of the world. This presented study gives an interesting insight into the indications for the same, in a limited area of Spain as an EU country.
This rather interesting retrospective observational study gives a complete overview of all modes of childbirth during a strict period, with the listed indications. Similar studies are not common in recent literature.
This study does not have a significant clinical contribution in terms of novelties, but the study according to the authors is not conceived that way. The research gives an interesting insight and possible further comparison with similar such studies of other geographical areas.
General concept comments:
The title is meaningful, concise, and also indicates the targeted issues that will be considered in the research.
The abstract is written according to the rules of writing abstracts, the only drawback that I would personally cite is the addition of the number of total births, i.e. the total number of women born in the observed period.
In the keywords, I would like to add a few more words that indicate the topic of research.
The introduction is concise, interesting and factually clearly written.
When considering the design of the study , the study itself is limited by using electrical data without the need for contact with patients, so that a certain percentage of Caesarean sections do not even have a clearly set indication, i.e. the same is not recorded. It has been recorded that there is a license of the ethics committee of the institution, the legitimacy of the same is questionable in my opinion, but I am not a legal expert. I think the same needs to be explained more clearly.
Clearly and concisely with a good description, the included and exclusion criteria of this research are listed.
The tabular and graphical representations of the Robson classification through the analyzed time period are well presented.
In the discussion in row 234. a comparison with the rest of Spain is given in detail, a detailed explanation of the increase in the incidence of Caesarean sections as well as the induction of childbirth with a potential effect on maternal health is given. This was previously concisely stated in the results.
In line 246. the results with the EU were well compared, the percentage and target countries to which they refer were absent.
In row 251. it was stated that the study in most cases did not clearly document the reason for induction of childbirth (as much as 25.7%). How do you explain the same? I think it needs to be supplemented with some presumed clarification.
The shortcomings of the study are scantily and simplified, I think there needs to be a more extensive and detailed clarification there.
The power of the study is explained simplistically, too briefly. It is stated that the very strength of the study is the number of samples, that is large, and they propose new ways of better analyzing.
In the conclusion of the research, it was stated that the authors believe that the criteria for induction of childbirth in first-born women should be revised, because they most often end in childbirth by caesarean section.
I believe that the discussion should explain in more detail by which guidelines the institution is guided in the work in induction of childbirth. I believe that it is necessary to emphasize and enter a comparison of this research with the literature data of other countries or other institutions within Spain, this is not mentioned in the paper.
Manuscript is clear, concise and interesting to the narrow circle of gynecologists and obstetricians, more precisely obstetricians subspecialists of fetal medicine and obstetrics.
The quoted references are recent, namely 17/29 or 58.62% and relate to the topic of the work.
The results of the manuscript can be reproduced based on the results given in the methods section, taking into account the basic beginning of the design of the study, a retrospective study.
Author Response
Response to Reviewer 4 Comments
Thank you for your very positive and constructive feedback on our manuscript.
We have considered all your comments and suggestions (and also the comments made by the other reviewer) attempting to improve/refine the original manuscript.
Below you will find a point-by-point response to your comments (in red).
Point 1: A brief summary:
The aims of this study were to analys assess levels and trends of birth by Cesarean sections in La Ribera University Hospital , Spain during 11 - years period, using the Robson classification, describe the indications for induction of labour and the causes of caesarean sections performed, also they examine the association between induction of labour and birth by Cesaren section.
The increase in the number and percentage of Caesarean sections is a major public health problem, with a trend of growth over the years in all countries of the world. This presented study gives an interesting insight into the indications for the same, in a limited area of Spain as an EU country.
This rather interesting retrospective observational study gives a complete overview of all modes of childbirth during a strict period, with the listed indications. Similar studies are not common in recent literature.
This study does not have a significant clinical contribution in terms of novelties, but the study according to the authors is not conceived that way. The research gives an interesting insight and possible further comparison with similar such studies of other geographical areas.
Response 1: Thank you for your comments. We believe that the study complies with the WHO requirements in that we publish the data in the recommended format to make results visible in order to be able to compare data from different countries and health systems. The purpose of capturing the results through the Robson classification is to establish a starting point to be able to establish strategies to improve caesarean section rates. Therefore, considering that our induction rates are the weakest point, we propose to analyse our local guidelines to reduce caesarean section rates.
Point 2: General concept comments:
The title is meaningful, concise, and also indicates the targeted issues that will be considered in the research.
The abstract is written according to the rules of writing abstracts, the only drawback that I would personally cite is the addition of the number of total births, i.e. the total number of women born in the observed period.
In the keywords, I would like to add a few more words that indicate the topic of research.
The introduction is concise, interesting and factually clearly written.
When considering the design of the study , the study itself is limited by using electrical data without the need for contact with patients, so that a certain percentage of Caesarean sections do not even have a clearly set indication, i.e. the same is not recorded. It has been recorded that there is a license of the ethics committee of the institution, the legitimacy of the same is questionable in my opinion, but I am not a legal expert. I think the same needs to be explained more clearly.
Response 2: Thank you for your comments. Regarding the ethical comment, the Declaration of Helsinki, in its point 32, indicates "There may be exceptional situations in which it will be impossible or impracticable to obtain consent for said research". In this situation, the research can only be carried out after being considered and approved by a research ethics committee: World Medical Association. Ethics Unit. Declaration of Helsinki 2007. www.wma.net/e/ethicsunit/helsinki.htm. This situation occurs in our case. We obtained the positive report from the Investigation Committee of our Hospital, as indicated in lines 92-96 and 367-372.
Regarding the comment on the need to contact the participants to complete missing data regarding the indication for cesarean section, it is recognized in the limitations section, being inherent to retrospective studies.
In relation to the total number of births, we are unsure that we understood the comment. The total number of births is specified in line 149. The RTGC analysis only includes the number of births rather than the number of newborns born. If any amendment requires to be performed on the manuscript, we would appreciate a clarification for this point.
Regarding the keywords, thank you for your input. We have now included the keywords: indications; onset of labour; mode of birth.
Point 3: Clearly and concisely with a good description, the included and exclusion criteria of this research are listed.
The tabular and graphical representations of the Robson classification through the analyzed time period are well presented.
In the discussion in row 234. a comparison with the rest of Spain is given in detail, a detailed explanation of the increase in the incidence of Caesarean sections as well as the induction of childbirth with a potential effect on maternal health is given. This was previously concisely stated in the results.
In line 246. the results with the EU were well compared, the percentage and target countries to which they refer were absent.
Response 3: Thank you for your input.
Point 4: In row 251. it was stated that the study in most cases did not clearly document the reason for induction of childbirth (as much as 25.7%). How do you explain the same? I think it needs to be supplemented with some presumed clarification.
Response 4: Thank you for your highlight. This is a typographical error. Instead of 25.7% it is 5.4% (line 311), since there were 363 women that received an induction and there was no clinical indication was documented in the clinical records out of a total of 6776 women receiving inductions, therefore 363/6776= 5.4%.
Point 5: The shortcomings of the study are scantily and simplified, I think there needs to be a more extensive and detailed clarification there. The power of the study is explained simplistically, too briefly. It is stated that the very strength of the study is the number of samples, that is large, and they propose new ways of better analyzing.
Response 5: Thank you for your comment. We understand that this refers to the limitations section. A new paragraph has been written in lines 330-336.
Point 6: In the conclusion of the research, it was stated that the authors believe that the criteria for induction of childbirth in first-born women should be revised, because they most often end in childbirth by caesarean section.
I believe that the discussion should explain in more detail by which guidelines the institution is guided in the work in induction of childbirth. I believe that it is necessary to emphasize and enter a comparison of this research with the literature data of other countries or other institutions within Spain, this is not mentioned in the paper.
Response 6: Thank you for your comment. Various comparisons with other studies have been added.
Point 7: Manuscript is clear, concise and interesting to the narrow circle of gynecologists and obstetricians, more precisely obstetricians subspecialists of fetal medicine and obstetrics.
The quoted references are recent, namely 17/29 or 58.62% and relate to the topic of the work.
The results of the manuscript can be reproduced based on the results given in the methods section, taking into account the basic beginning of the design of the study, a retrospective study.
Response 7: Thank you for your highlight.
Point 8: Should note the analysis done to arrive at the p-value(s). Tables should be internally consistent in the number of decimal places. In general, it does not help readers to report percentages to more than one decimal place.
It also does not help readers to report odds ratios or their CIs to 3 decimal places (especially when the OR is only reported to 2).
Response 8: Thank you for your highlight. Amended in all sections of the manuscript.

Round 2
Reviewer 1 Report
Review of Healthcare-23028891-v2 (I think this is the right number)
Thanks to the authors for the improvements in their paper.
A remark on English usage (trying to be helpful): The main use of “indication” (and usually “indicated”) in medical writing is “legitimate reason for” (at least by current medical standards). In these data, clearly some “indications,” such as “favorable cervix” are not legitimate reasons (though they may once have been so—my ignorance), but are apparently the only justification given. The authors also use the word “indicate” where they mean “note” or “mention,” possibly a carry-over from Spanish. I will try to comment on these places.
Statistical comment: the Chi-squared and Kruskal-Wallis tests do not test trend, but associations of categorical or continuous variables with a categorical variable. I gather that the authors performed these tests, then remarked on increases or decreases if the numbers seemed to have a trend. In general, formal trend tests would be better, but I am not insisting on them.
Also, when you have large numbers, as you do here, small differences can be “statistically significant” without being important.
General comment: Classifications of this sort (RTGCS) can be very helpful, but suggested c-section rates for each category are suggestions (not laws), and optimal rates may change by time and place. One should not thing of them as “revealed truth.”
Thinking causally, we expect maternal nationality, maternal age, calendar year, previous CS, number of fetuses, number of prior pregnancies, presentation to “cause” the RTGCS group (especially since some of these factors are part of the definitions of the groups. In other words, women show up for labor with a specific set of characteristics (age, parity, previous c-section status, intention to have an elective c-section, etc.) and during the labor and delivery they end up as c-section, induction without c-section, or neither. Analyses that condition on the group to look at the causal factors, such as that in lines 180-190 are thus backwards. Nonetheless, I will let it stand, because it’s still interesting, and I don’t think we will get anything better. Indeed, it would be best to look at the c-sections as a subset of all deliveries during this time and look at the factors related to trends in c-section.
Specific comments
Line comment
1 29 Please change “models adjustment” to “regression.”
2 34 “maintained an upward trend (16.5%)”—Please change “maintained” to “showed,” since there are times when the rate went down. I gather that the 16.5% is the last rate – the first rate. I think this would be clearer if you wrote “showed an upward trend, from 22.1% to 38.7%” (whatever the numbers are—I’m having trouble reading the graph).
3 65 Please insert “as” between “established” and “a.”
4 69 Please insert “of” between “induction” and “labor.”
5 136 Please say “nonnormal,” rather than “abnormal.”
6 151 Please insert “were” before “induced.”
7 163-164 I don’t think it is necessary to include “This difference… “, though it would be fine to include the p-value in parentheses at the end of the previous sentence.
8 167 I think you mean “one previous pregnancy.”
9 168 Similarly, “two or more previous pregnancies”--This comment and the previous one may apply in other places.
10 169-170 Again, “and this difference…” is not necessary, except that this time it is not clear to what the p-value refers. It could be the 2x2 table comparing rates at beginning and end for one previous pregnancy to rates for more than one previous pregnancy, or it could just refer to the change in the “more than one” group. I think that the only way to make this truly clear is to put in p-values for both clauses.
11 170 In your future, you may be interested in the proportions of women in the extremes of maternal age (e.g., very young, however that is defined these days, and over 35), rather than the means.
12 172 Probably “of previous births”
13 Table 2 The sum of the numbers in the first box is 20481, not 20578. There is no mention anywhere of the fact that some patients had missing data (other than missing indication for induction). YOU NEED TO DISCUSS MISSING DATA AND HOW YOU HANDLED THEM.
14 188 Please specify the group(s) to which the increase from 31 to 35 applies. Still groups 5 and 7? The several groups mentioned? The wording might imply “overall,” but the overall increase is just 1.1 years.
15 Table 3 The header of the fourth column should be a bit more descriptive, such as “proportional” or “relative” group size.
16 214 This is a case where I think that “noted” works better than “indicated.”
17 218 “favorable cervix” is so close in numbers to these 2 that I think you have to include it here. Obviously, this is not a real indication for induction, and presumably accounts for a lot of c-sections. You could presumably get the actual number from your data.
18 223 “labour induction was indicated” would probably be better as “labour was induced.”
19 231 I would like to see failed induction and obstructed labour added here. If you think that CPD and noncephalic presentation should also be in the list, I have no objections.
20 246 Is the number really higher, or is it just the fraction/proportion? After all, the total number of births decreased considerably. Based on the supplementary table, the actual number went down.
21 247 “mentioned” or “noted” is probably better than “indicated” at the end of the sentence.
22 259 Drop “in depth.”
23 276 “mentioned” is probably better than “indicated.”
24 308-311 “it is of great….documented”--This is awkward. Please reword. Suggestion: The large fraction of women in our study who received IOLs that were not clinically indicated (9.3% “favourable cervix”) or for which the indication for IOL was not documented (5.4%) is of great concern.
25 338-340 I don’t know what this sentence means.
26 340-341 Does this mean that they counted in 2 different ways and got the same answer? While that is somewhat reassuring, it is not in itself evidence of high quality data. See remarks under Table 2.
27 341-342 I have no idea what this sentence is supposed to mean.
Author Response
Response to Reviewer 1 Comments Round 2
Thank you for your positive and constructive feedback on our manuscript.
We have considered all your comments and suggestions and also the comments made by the others reviewers attempting to improve/refine the original manuscript.
Below you will find a point-by-point response to your comments (in red).
Point 1: Thanks to the authors for the improvements in their paper.
A remark on English usage (trying to be helpful): The main use of “indication” (and usually “indicated”) in medical writing is “legitimate reason for” (at least by current medical standards). In these data, clearly some “indications,” such as “favorable cervix” are not legitimate reasons (though they may once have been so—my ignorance), but are apparently the only justification given. The authors also use the word “indicate” where they mean “note” or “mention,” possibly a carry-over from Spanish. I will try to comment on these places.
Response 1: Thank you for your help.
Point 2: Statistical comment: the Chi-squared and Kruskal-Wallis tests do not test trend, but associations of categorical or continuous variables with a categorical variable. I gather that the authors performed these tests, then remarked on increases or decreases if the numbers seemed to have a trend. In general, formal trend tests would be better, but I am not insisting on them.
Response 2: Thank you for your understanding. Indeed, our intention is to try to perform a descriptive analysis, as done when using the WHO's Robson classification, which describes "trends" in the form of percentages over the years of follow-up. In any case, to avoid misinterpretations, we have decided to change the term 'trend' to 'distribution' in scientific English (lines 22, 77, and 159).
Point 3: Also, when you have large numbers, as you do here, small differences can be “statistically significant” without being important.
Response 3: We agree with the argument that when dealing with large numbers, small differences can be statistically significant without necessarily being important or meaningful. In statistical analysis, it is important to consider not only the statistical significance but also the practical significance of the results. Therefore, it is crucial to interpret the results in the context of the research question and to consider the effect size, clinical relevance, and potential impact of the findings. We understand that most of the observed statistical differences have been commented on and put into context. If not, please indicate which ones, if any, you think require further justification so that we can address them accordingly.
Point 4: General comment: Classifications of this sort (RTGCS) can be very helpful, but suggested c-section rates for each category are suggestions (not laws), and optimal rates may change by time and place. One should not thing of them as “revealed truth.”
Response 4: We completely agree with this comment. Optimal rates may vary depending on various factors, including local practices, resources, and cultural preferences. Therefore, it is important to use these classifications as tools for informed decision-making and to consider each individual case on its own merits, taking into account the risks and benefits for both the mother and the baby.
Point 5: Thinking causally, we expect maternal nationality, maternal age, calendar year, previous CS, number of fetuses, number of prior pregnancies, presentation to “cause” the RTGCS group (especially since some of these factors are part of the definitions of the groups. In other words, women show up for labor with a specific set of characteristics (age, parity, previous c-section status, intention to have an elective c-section, etc.) and during the labor and delivery they end up as c-section, induction without c-section, or neither. Analyses that condition on the group to look at the causal factors, such as that in lines 180-190 are thus backwards. Nonetheless, I will let it stand, because it’s still interesting, and I don’t think we will get anything better. Indeed, it would be best to look at the c-sections as a subset of all deliveries during this time and look at the factors related to trends in c-section.
Response 5: Thank you for your comment. We agree that there are many factors that may influence the RTGCS group classification, and we have taken these factors into consideration in our analysis. However, we acknowledge that our analysis is limited by the data available to us and the specific context of our study. We appreciate your suggestion to consider analysing c-section rates as a subset of all deliveries and examining factors related to trends in c-section. We will keep this in mind for future research.
Specific comments
Line comment
1 29 Please change “models adjustment” to “regression.”
Response 1: Thank you. Amended in line 29.
2 34 “maintained an upward trend (16.5%)”—Please change “maintained” to “showed,” since there are times when the rate went down. I gather that the 16.5% is the last rate – the first rate. I think this would be clearer if you wrote “showed an upward trend, from 22.1% to 38.7%” (whatever the numbers are—I’m having trouble reading the graph).
Response 2: Thank you. Amended in line 33-34 and updated the percentages.
3 65 Please insert “as” between “established” and “a.”
Response 3: Thank you. Amended in line 64.
4 69 Please insert “of” between “induction” and “labor.”
Response 4: Thank you. Amended in line 68.
5 136 Please say “nonnormal,” rather than “abnormal.”
Response 5: Thank you. Amended in line 135.
6 151 Please insert “were” before “induced.”
Response 6: Thank you. Amended in line 150.
7 163-164 I don’t think it is necessary to include “This difference… “, though it would be fine to include the p-value in parentheses at the end of the previous sentence.
Response 7: Thank you. Amended in lines 163-164.
8 167 I think you mean “one previous pregnancy.”
Response 8: Thank you. Amended in line 166.
9 168 Similarly, “two or more previous pregnancies”--This comment and the previous one may apply in other places.
Response 9: Thank you. Amended in line 167 and in table 2.
10 169-170 Again, “and this difference…” is not necessary, except that this time it is not clear to what the p-value refers. It could be the 2x2 table comparing rates at beginning and end for one previous pregnancy to rates for more than one previous pregnancy, or it could just refer to the change in the “more than one” group. I think that the only way to make this truly clear is to put in p-values for both clauses.
Response 10: Thank you. Amended in line 168.
11 170 In your future, you may be interested in the proportions of women in the extremes of maternal age (e.g., very young, however that is defined these days, and over 35), rather than the means.
Response 11: Thank you for your comment. We agree that there are many factors that may influence on the perinatal outcomes including the maternal age as described in previous studies. We will keep this in mind for future research.
12 172 Probably “of previous births”
Response 12: Thank you for your comment. The variable births correspond to the number of births attended during the periods. However, as already considered, we did include the term "previous pregnancy” in comment 9.
13 Table 2 The sum of the numbers in the first box is 20481, not 20578. There is no mention anywhere of the fact that some patients had missing data (other than missing indication for induction). YOU NEED TO DISCUSS MISSING DATA AND HOW YOU HANDLED THEM.
Response 13: Thank you for your highlight. We have performed a new analysis with the complete database (20,578), and the results have been adjusted with the updated values. Please note that the previous analysis was performed on a subset of the data due to the selection of a filter related to caesarean section indications, which resulted in 97 fewer cases being analysed. However, the new analysis was carried out on the entire dataset, and the results have been updated accordingly in Table 2, and lines 166-174.
14 188 Please specify the group(s) to which the increase from 31 to 35 applies. Still groups 5 and 7? The several groups mentioned? The wording might imply “overall,” but the overall increase is just 1.1 years.
Response 14: Thank you for your comment. The increase in maternal age from 31 to 35 applies to group 5 and 7. We have amended line 186 in order to clarify this.
15 Table 3 The header of the fourth column should be a bit more descriptive, such as “proportional” or “relative” group size.
Response 15: Thank you for your comment. We agree with the comment, but the RTGCS uses these labels. If it is not deemed necessary, we would prefer to keep them as they are.
16 214 This is a case where I think that “noted” works better than “indicated.”
Response 16: Thank you for your comment. This now been amended in lines 212.
17 218 “favorable cervix” is so close in numbers to these 2 that I think you have to include it here. Obviously, this is not a real indication for induction, and presumably accounts for a lot of c-sections. You could presumably get the actual number from your data.
Response 17: Thank you for your comment. Exactly, as this is not a real indication for induction of labor we preferred not to describe this as such. However, we have highlighted and discussed this finding in lines 303-309.
18 223 “labour induction was indicated” would probably be better as “labour was induced.”
Response 18: Thank you for your help. Amended in line 221.
19 231 I would like to see failed induction and obstructed labour added here. If you think that CPD and noncephalic presentation should also be in the list, I have no objections.
Response 19: Thank you for your highlight. We have now added this in lines 228-229.
20 246 Is the number really higher, or is it just the fraction/proportion? After all, the total number of births decreased considerably. Based on the supplementary table, the actual number went down.
Response 20: Thank you. We are unsure we fully understood the comment. As discussed in the previous review, the overall yearly number of births in our study decreases over the study years according to the decrease in the national number of births. However, as shown in Figure 1 the relative contribution of groups 2 and 4 increases. Please, clarify the comment if changes are required.
21 247 “mentioned” or “noted” is probably better than “indicated” at the end of the sentence.
Response 21: Thank you for your help. Amended in line 245-246. We have used the term "performed" as this might be more appropriate in this context.
22 259 Drop “in depth.”
Response 22: Thank you for your help. Amended in line 258.
23 276 “mentioned” is probably better than “indicated.”
Response 23: Thank you for your help. Amended in line 275. In the same context as before, we prefer to use "performed".
24 308-311 “it is of great….documented”--This is awkward. Please reword. Suggestion: The large fraction of women in our study who received IOLs that were not clinically indicated (9.3% “favourable cervix”) or for which the indication for IOL was not documented (5.4%) is of great concern.
Response 24: Thank you for your help. We agree with the suggestion and we have amended in lines 307-309.
25 338-340 I don’t know what this sentence means.
Response 25: Thank you for your comment. In order to clarify the idea, a new paragraph has been written in lines 338-340 as follows:
Data quality is confirmed as the research team followed the RTGCS directives to report the data and the total number of CS and overall number of births reported in this study is identical to those reported by the hospital.
26 340-341 Does this mean that they counted in 2 different ways and got the same answer? While that is somewhat reassuring, it is not in itself evidence of high quality data. See remarks under Table 2.
Response 26: Thank you for your comment. We think that with the new previous paragraph this has now been clarified.
27 341-342 I have no idea what this sentence is supposed to mean.
Response 27: Thank you for your comment. A new paragraph has been rewritten to clarify the sentence in lines 340-343 as follows:
In addition, as stated by the RTGCS data quality reporting guidelines, the number of singleton pregnancies in a transverse position (group 9) should be less than 1% as it is reported in this study.
